# ROYAL SOCIETY
# OPEN SCIENCE

mathematical modelling/differential equations/ health and disease and epidemiology

Coronavirus, statistical analysis, extrapolation, parameter estimation, pandemic spreading

**Author for correspondence:**
M. Kröger
e-mail: mk@mat.ethz.ch

# Verification of the accuracy of the SIR model in forecasting based on the improved SIR model with a constant ratio of recovery to infection rate by comparing with monitored second wave data

M. Kröger[1] and R. Schlickeiser[2,3]

[1]Department of Materials, Polymer Physics, ETH Zurich, Zurich CH-8093, Switzerland [2]Institut für Theoretische Physik, Lehrstuhl IV: Weltraum- und Astrophysik, Ruhr-Universität Bochum, Bochum 44780, Germany [3]Institut für Theoretische Physik und Astrophysik, Christian-Albrechts-Universität zu Kiel, Leibnizstr. 15, Kiel D-24118, Germany

MK, 0000-0003-1402-6714; RS, 0000-0003-3171-5079

The temporal evolution of second and subsequent waves of epidemics such as Covid-19 is investigated. Analytic expressions for the peak time and asymptotic behaviours, early doubling time, late half decay time, and a half-early peak law, characterizing the dynamical evolution of number of cases and fatalities, are derived, where the pandemic evolution exhibiting multiple waves is described by the semi-time SIR model. The asymmetry of the epidemic wave and its exponential tail are affected by the initial conditions, a feature that has no analogue in the all-time SIR model. Our analysis reveals that the immunity is very strongly increasing in several countries during the second Covid-19 wave. Wave-specific SIR parameters describing infection and recovery rates we find to behave in a similar fashion. Still, an apparently moderate change of their ratio can have significant consequences. As we show, the probability of an additional wave is however low in several countries due to the fraction of immune inhabitants at the end of the second wave, irrespective of the ongoing vaccination efforts. We compare with alternate approaches and data available at the time of submission. Most recent data serves to demonstrate the successful forecast and high accuracy of the SIR model in predicting the evolution of pandemic outbreaks as long as the assumption underlying our analysis, an unchanged

situation of the distribution of variants of concern and the fatality fraction, do not change dramatically during a wave. With the rise of the $\alpha$ variant at the time of submission the second wave did not terminate in some countries, giving rise to a superposition of waves that is not treated by the present contribution.

# 1. Introduction

Currently, many countries over the world have to cope with handling the subsequent (second, third …) wave outbreaks of Covid-19 pandemic. The onset of a subsequent wave is most probably caused by a sudden mutation of the virus resulting in an effective (about 40%) increase of the infection rate. Of high medical and economic interest are reliable predictions on the duration, peak days and total amplitudes of both the number of infections and the number of fatalities which are closely related to the maximum need for breathing apparati in hospitals in order not to overburden the medical clinics capacities and to avoid fateful triage decisions.

In the past reasonably accurate (within 50%) predictions on the first Covid-19 wave have been made [1–4] adopting the Gaussian distribution for the daily rate of new cases and the corresponding cumulative numbers (including both infections and fatalities) for many countries in the world with well-monitored case rates. By modelling simultaneously both infection and fatality rates the dark number of infections and the degree of herd immunity from the first wave has been determined in several countries [5] by adopting a mortality rate of 0.5%, i.e. one out of 200 infected persons ultimately dies from Covid-19. While the Gauss distribution can be justified both from the central limit theorem of statistics [1] and for early times until the peak time from the susceptible-infectious-recovered/removed (SIR) epidemic model [6–9], it is less accurate at late times of the wave evolution as compared to monitored data [10].

The SIR model [6,7] describes the time evolution of infectious diseases in human populations, and is the simplest and most fundamental of the compartmental models and its variations. It had been solved numerically using various approaches, including Monte Carlo methods, wavelets, fuzzy control, deep learning etc. [11–31] and approximate solutions had been proposed [32–34]. Our recent work [35,36] improved the analytical modelling of epidemics based on the well-established SIR epidemic model invented nearly 100 years ago.

There are two variants of implementing the SIR model, as described in detail previously [35,36]. The all-time SIR model must use an initial condition that is compatible with the SIR equations of change, which allows it to 'predict' both the past and present consistently. The semi-time SIR model can be used with any initial condition, but makes predictions only about the future. Often the second variant is used without noticing the inconsistency, while the inconsistency had already been noted by Kendall [6]. Assuming a constant ratio between infection and recovery rates, the all-time SIR model does predict a single wave only, while the semi-time SIR model we are going to use in this work allows us to model many waves with their own specific parameters, that capture only the future development in each case.

For both variants, the considered population of $N \gg 1$ persons is assigned to the three compartments $s$ (susceptible), $i$ (infectious) or $r$ (recovered/removed). Persons from the population may progress with time between these compartments with given infection ($a(t)$) and recovery rates ($\mu(t)$) which in general vary differently with time. As demonstrated before [35], it is convenient to introduce with $I(t) = i(t)/N$, $S(t) = s(t)/N$ and $R(t) = r(t)/N$ the infected, susceptible and recovered/removed fractions of persons involved in the infection at time $t$, with the sum requirement $I(t) + S(t) + R(t) = 1$. Moreover, if $t_{0,n}$ denotes starting time of the $n$th wave with the initial conditions $I(t_{0,n}) = \eta$ and $S(t_{0,n}) = 1 - \eta$, it is also appropriate to introduce the dimensionless reduced time variable

$$\tau(t) = \int_{t_{0,n}}^{t} \mathrm{d}\xi a(\xi), \tag{1.1}$$

accounting for arbitrary but given time-dependent infection rates $a(t)$. With the introduction of the reduced time (1.1) the following analysis includes and addresses the effects of non-pharmaceutical interventions (NPIs). These affect the infection rate $a(t)$ by lowering it to lower values providing substantial differences to the simple linear relation $\tau = a_0 t$ of the reduced time as a function of the real time $t$ in case of an unchanged initial infection rate $a_0$. This has been described and illustrated in detail in §§2.2 and 2.3 of earlier work [10].

For the special and important case of a time-independent ratio $K(t) = \mu(t)/a(t) = k = \mathrm{const.}$, analytical results of the SIR model have been recently derived [35,36] both for the all-time and semi-time cases appropriate for the first and subsequent wave evolution, respectively. For a growing epidemics outburst,

exhibiting a peak at some time past $t_{0,n}$ in the future, the constant ratio $k$ has to be smaller than $1 - 2\eta$, where as noted before $\eta = I(t_{0,n})$ denotes the infected fraction at the starting time $t_{0,n}$ of the $n$th wave. The requirement $k < 1 - 2\eta$ corresponds to the initial infection rate $a_0$ at time $t_{0,n}$ being larger than the initial recovery rate $\mu_0$, both in units of days$^{-1}$. Alternatively, for initial ratios $k > 1 - 2\eta$, the wave amplitude is purely decaying from its maximum at $t_{0,n}$ without exhibiting a wave phenomenon [36]. Here, the time $t_{0,n}$ refers to the observing time when the onset of a new $n$th wave is recognized by the monitoring of case rates.

It has been demonstrated [36] that the two parameters $k$ and $\eta$ determine the full evolution of each subsequent wave in reduced time (1.1) including in particular the effects of NPIs which determine the relation between reduced time and real time for non-stationary infection rates. Moreover, it is important to emphasize that in the peak case where $k < 1 - 2\eta$ the SIR model not only provides a causal connection between the early (before the peak) and late time development of the pandemic wave, but that the parameters $k$ and $\eta$ obtained from fitting the early time evolution also determine the final and maximum values of the cumulative and daily case rates.

Below, we will derive simple analytic expressions for all measurable amounts of cases and fatalities during a pandemic evolution described by the semi-time SIR model, that share all relevant features with the exact solution of the semi-time SIR model, including time and position of the peak of daily new infections, as well as the asymptotic behaviours at small and large times. The expressions are so precise that they can be used instead of a numerical solution of the SIR model. The advantage of an analytical expression is obvious, as it allows us to quickly determine the SIR parameters from the measured data well ahead of the peak time, and thus allows for predictions that serve as a prerequisite to make decisions. We apply the approach to eight different countries from different continents. We begin by summarizing the exact features of the semi-time SIR model, stating the approximants for the reliably measurable quantities, and collect all the derivations of the new approximant in an appendix. The occurrences of a clearly separated (in time) second wave in these countries provide very suitable data sets to verify the forecasts of the SIR model which is one important motivation for this study here.

Throughout this work, we ignore the effect of vaccination campaigns on the temporal evolution which we investigate elsewhere [37]. In that reference, it is demonstrated that the generalization to solutions with vaccinations relies heavily on very accurate analytical solutions of the SIR model in order to interpolate especially at small values of the ratio of vaccination to infection rates. It is precisely the purpose of this study here to provide such accurate solutions of the SIR model. We furthermore assume an unchanged situation of the distribution of variants of concern and the fatality fraction during a single wave.

In order for the interested reader to comprehend why our analysis is based on predictions made in January 2021, we note that the original version of this manuscript had been received January 16, 2021 by the Royal Society. It was passed over internally to Royal Society Open Science in April 2021. The original forecast was based on data collected on January 11, 2021. For the purpose of this study, the original forecast has not been revised using later data, but later data was added to compare with the original forecast. The original preprint of this manuscript entitled 'Forecast made on January 11, 2021 for the second Covid-19 wave based on the improved SIR model with a constant ratio of recovery to infection rate' is available since January 21, 2021 at preprints.org. The title had been modified as the forecast is not a forecast anymore at the time of publication of this manuscript.

## 2. SIR-model results

### 2.1. Exact results

The exact solution of the SIR model with a constant ratio of the recovery to infection rate in the semi-time case [36] (hereafter referred to as KS-SIR model) is given by

$$\tau = \int_{\eta}^{J} \frac{\mathrm{d}y}{n(y)} \tag{2.1a}$$

and

$$n(y) = (1 - y)[y + k\varepsilon + k\ln(1 - y)], \tag{2.1b}$$

with $\eta = 1 - \mathrm{e}^{-\varepsilon}$, $\tau$ given by equation (1.1), and $J = 1 - S$ denoting the cumulative fraction of new cases. Differentiating equation (2.1a) with respect to $\tau$ yields for the corresponding rate of new cases

$$j(\tau) = \frac{\mathrm{d}J(\tau)}{\mathrm{d}\tau} = (1 - J)[J + k\varepsilon + k\ln(1 - J)] = n(J). \tag{2.2}$$

As shown before [36] without explicit inversion of equation (2.1), equations (2.1) and (2.2) yield for the final cumulative number fraction of infected persons $J_\infty = \lim_{\tau \to \infty} J(\tau)$ and for the maximum rate of new infections $j_{max}$ occurring at $J_0$ that

$$J_\infty = 1 + kW_0(\alpha), \quad \alpha = -\frac{(1-\eta)e^{-1/k}}{k} \tag{2.3}$$

$$J_0 = 1 + \frac{k}{2}W_{-1}(\alpha_0), \quad \alpha_0 = \frac{2\alpha}{e} \tag{2.4}$$

$$j_{max} = n(J_0) = (1 - J_0)(1 - J_0 - k) \tag{2.5}$$

$$= \frac{k^2}{4}\left([1 + W_{-1}(\alpha_0)]^2 - 1\right), \tag{2.6}$$

in terms of the principal ($W_0$) and non-principal ($W_{-1}$) solutions of Lambert's equation [35], the well-known Lambert's functions. The asymmetry of $j(\tau)$ about its peak time is encoded in a difference between $J_0$ and $J_\infty/2$. We emphasize again that the integral quantities $J_\infty$ and $J_0$ refer to the pandemic evolution in reduced time so it includes the effects of NPIs which determine the relation between real and reduced time.

## 2.2. Constrained second-order polynomial approximation

We approximately invert the solution (2.1) by using the constrained Taylor-expansions of the reciprocal integrand

$$n(y) \simeq j_c(y) = \sum_{i=0}^{2} c_i(y - \eta)^i \tag{2.7}$$

$$= c_0 + c_1(y - \eta) + c_2(y - \eta)^2$$

and

$$n(y) \simeq j_d(y) = \sum_{i=1}^{2} d_i(J_\infty - y)^i \tag{2.8}$$

$$= d_1(J_\infty - y) + d_2(J_\infty - y)^2$$

about $y_c = \eta$ and $y_d = J_\infty$ up to second order, respectively. Here $c_0$, $c_1$ and $d_1$ are the respectively positively valued Taylor expansion coefficients given by

$$c_0 = (1 - \eta)\eta \tag{2.9}$$

$$c_1 = 1 - k - 2\eta \tag{2.10}$$

and

$$d_1 = J_\infty - (1 - k). \tag{2.11}$$

Hereby, constrained expansion refers to choosing the second-order expansion coefficients $c_2$ and $d_2$ such that the respective expansion evaluated at $y = J_0$ yields the maximum value of the daily case rate, i.e.

$$j_c(y = J_0) = j_d(y = J_0) = j_{max}, \tag{2.12}$$

with $j_{max}$ from equation (2.6). The requirement (2.12) then readily provides

$$c_2 = \frac{j_{max} - c_0 - c_1(J_0 - \eta)}{(J_0 - \eta)^2} \tag{2.13}$$

and

$$d_2 = \frac{j_{max} - d_1(J_\infty - J_0)}{(J_\infty - J_0)^2}. \tag{2.14}$$

While $c_2$ is negative for all $k < 1 - 2\eta$, the coefficient $d_2$ may have either sign. It is important to realize that all quantities (2.3)–(2.6) and coefficients (2.9)–(2.11) as well as (2.13) and (2.14) are solely determined by the two basic KS-SIR model parameters $k$ and $\eta$.

According to equations (2.7) and (2.8), we have thus constructed the approximation for the daily rate $j(J)$ as a function of the cumulative number

$$j(J) \simeq \begin{cases} c_0 + c_1(J - \eta) + c_2(J - \eta)^2 & \text{for } J \leq J_0 \\ d_1(J_\infty - J) + d_2(J_\infty - J)^2 & \text{for } J \geq J_0, \end{cases} \quad (2.15)$$

which is continuous at $J = J_0$ where $j(J_0) = j_{max}$ attains its peak value. As shown in the following by demanding that our approximation (2.15) attains the exact maximum value at $J = J_0$ and a vanishing final value at $J = J_\infty$ we obtain a very high agreement between the approximated analytical and the exact numerical pandemic evolution as a function of the reduced time $\tau$.

As we show next, inserting the approximation (2.15) then allows the analytical inversion of the solution (2.1) providing $J(\tau)$ as a function of the reduced time $\tau$ which then can be used either in equation (2.15) or equation (2.2) to infer also the rate $j(\tau)$ as a function of reduced time. The remaining SIR quantities are then obtained from $J(\tau)$ as well via $S(\tau) = 1 - J(\tau)$, $I(\tau) = J(\tau) + k\varepsilon + k\ln[1 - J(\tau)]$ and $R(\tau) = 1 - S(\tau) - I(\tau) = -k[\varepsilon + \ln(1 - J(\tau))]$.

## 2.3. Cumulative number and rate by exact inversion of the approximant

While we have expressed $\tau$ in terms of $J$ above, for all practical purposes one is interested in the reverse relationship, the time $\tau$-dependent behaviour of $J$ and also $j$. With the approximations (2.7) and (2.8) we obtain for the solution (2.1)

$$\tau \simeq \begin{cases} \int_\eta^J \frac{dy}{j_c(y)} & \text{for } J \leq J_0 \\ \int_\eta^{J_0} \frac{dy}{j_c(y)} + \int_{J_0}^J \frac{dy}{j_d(y)} & \text{for } J \geq J_0. \end{cases} \quad (2.16)$$

Introducing the peak time scale

$$\tau_m = \int_\eta^{J_0} \frac{dy}{j_c(y)}, \quad (2.17)$$

corresponding to $J = J_0$ we may write equation (2.16) as

$$\tau \simeq \begin{cases} \tau_m - \int_J^{J_0} \frac{dy}{j_c(y)} & \text{for } \tau \leq \tau_m \\ \tau_m + \int_{J_0}^J \frac{dy}{j_d(y)} & \text{for } \tau \geq \tau_m \end{cases} \quad (2.18)$$

In appendix A, the integrals appearing in equations (2.17) and (2.18) are calculated leading to

$$J(\tau) \simeq \begin{cases} \eta + \dfrac{J_0 - \eta}{1 + \sqrt{\frac{j_{max}}{c_0}} \frac{\sinh[c_3(\tau_m - \tau)]}{\sinh(c_3 \tau)}} & \text{for } \tau \leq \tau_m \\ J_\infty - \dfrac{J_\infty - J_0}{\frac{j_{max}}{d_1(J_\infty - J_0)}[e^{d_1(\tau - \tau_m)} - 1] + 1} & \text{for } \tau \geq \tau_m, \end{cases} \quad (2.19)$$

with the dimensionless peak time

$$\tau_m = \frac{1}{c_3} \text{artanh} \frac{2c_3}{c_1 + (2c_0/(J_0 - \eta))}, \quad (2.20)$$

and the abbreviation

$$c_3 = \sqrt{(c_1/2)^2 - c_0 c_2}. \quad (2.21)$$

Note that equation (2.19) obviously exhibits the correct extremal behaviours, $J(0) = \eta$, $J(\tau_m) = J_0$ and $\lim_{\tau \to \infty} J(\tau) = J_\infty$.

Inserting the cumulative number $J(\tau)$ from equation (2.19) into equation (2.15), or alternatively, from $dJ/d\tau$, we obtain for the time dependence of the corresponding reduced rate (2.15)

$$\frac{j(\tau)}{j_{max}} = \begin{cases} \left( \dfrac{\sinh(c_3 \tau_m)}{\sinh(c_3 \tau) + \sqrt{j_{max}/c_0} \sinh[c_3(\tau_m - \tau)]} \right)^2 & \text{for } \tau \leq \tau_m \\ \dfrac{e^{d_1(\tau - \tau_m)}}{(1 + (j_{max}/d_1(J_\infty - J_0))[e^{d_1(\tau - \tau_m)} - 1])^2} & \text{for } \tau \geq \tau_m. \end{cases} \quad (2.22)$$

The solutions (2.22) correctly reduce to $j_{max}$ for $\tau = \tau_m$. Moreover, $j(0) = c_0 = \eta(1 - \eta)$ and $\lim_{\tau \to \infty} j(\tau) = 0$ are obviously correctly captured. Details of the calculations leading to equation (2.22) are also collected in appendix A. The relative errors of the approximants are reported in appendix C.

## 2.4. Early and late reduced time evolution

For very late reduced time $\tau \gg \tau_m$ the rate $j(\tau)$, second case in equation (2.22), approaches the decreasing exponential function in reduced time

$$
\begin{aligned}
j(\tau \gg \tau_m) &\simeq \frac{d_1^2 (J_\infty - J_0)^2}{j_{max}} e^{-d_1(\tau - \tau_m)} \\
&= \frac{[J_\infty - (1-k)]^2 (J_\infty - j_0)^2}{(1 - J_0)(1 - k - J_0)} e^{-[J_\infty - (1-k)](\tau - \tau_m)},
\end{aligned}
\tag{2.23}
$$

with the decay half-reduced time [38]

$$
\tau_{1/2} = \frac{\ln 2}{J_\infty - (1-k)} \approx \frac{0.693}{J_\infty - (1-k)},
\tag{2.24}
$$

defined by $j(\tau + \tau_{1/2}) = j(\tau)/2$. Similarly, for very early reduced time $\tau \ll \tau_m$, provided such a regime exists, as its pronounced appearance depends on the values for $\eta$ and $k$, the rate $j(\tau)$, first case in equation (2.22), is an increasing exponential function in reduced time with the doubling time

$$
\tau_2 = \frac{\ln \sqrt{2}}{c_3},
\tag{2.25}
$$

defined by $j(\tau + \tau_2) = 2j(\tau)$.

## 2.5. Half-early-peak law

The early differential rate exhibits a very interesting feature referred to here as the half-early-peak law. According to equation (2.22), the rate at half of the peak time $j_{1/2} = j(\tau_m/2)$ is given by

$$
\frac{j_{1/2}}{j_{max}} = \frac{4c_0 \cosh^2(c_3 \tau_m/2)}{(\sqrt{c_0} + \sqrt{j_{max}})^2} = \frac{4j(0) \cosh^2(c_3 \tau_m/2)}{(\sqrt{j(0)} + \sqrt{j_{max}})^2}.
\tag{2.26}
$$

The corresponding cumulative half-early-peak law follows from the first case of equation (2.19) as

$$
J_{1/2} = J\left(\frac{\tau_m}{2}\right) = \eta + \frac{J_0 - \eta}{1 + \sqrt{j_{max}/c_0}}
\tag{2.27}
$$

or

$$
1 + \sqrt{\frac{j_{max}}{c_0}} = \frac{J_0 - \eta}{J_{1/2} - \eta}.
\tag{2.28}
$$

With this equation the half-early-peak law (2.26) reads

$$
\begin{aligned}
\frac{j_{1/2}}{j_{max}} &= \left[\frac{2(J_{1/2} - \eta) \cosh(c_3 \tau_m/2)}{J_0 - \eta}\right]^2 \\
&= \left[\frac{2[J_{1/2} - J(0)] \cosh(c_3 \tau_m/2)}{J_0 - J(0)}\right]^2.
\end{aligned}
\tag{2.29}
$$

In case of temporal wave distributions with an apparent peak the half-early-peak law (2.29) provides an important test for the derived parameters of the wave as it relates directly the monitored quantities $J(0) = J(\tau = 0)$, $J_0 = J(\tau_m)$, $j_{1/2}$, $j_{max}$ and $c_3 \tau_m/2$, where the latter can also be written in terms of the ratio between peak time $\tau_m$ and early doubling time $\tau_2$ as $c_3 \tau_m/2 = \ln(2^{1/4}) \tau_m/\tau_2 \approx \frac{1}{6} \tau_m/\tau_2$.

## 2.6. Reintroducing dimensions: real time evolution

The rates (2.22) and the cumulative number (2.19) refer to the relative time $\tau$ defined in equation (1.1), whereas the monitored data refer to real time $t$. We therefore adopt, well justified for the initial phase of any new emergent wave, a constant infection rate $a(t) = a_0$ throughout so that $\tau = a_0(t - t_0)$, where for ease of exposition we drop the index $n$ and simply write $t_0 = t_{0,n}$. We then obtain in real time for the peak time

$$
t_m = t_0 + \frac{\tau_m}{a_0}.
\tag{2.30}
$$

Likewise, the pandemic time evolutions in real time follow readily from

$$J(t) = J(\tau = a_0(t - t_0))$$

and

$$\dot{J}(t) = a_0 j(\tau = a_0(t - t_0)),$$

(2.31)

and are written down in appendix B. Likewise, the doubling time at early times $t \ll t_m$ and the decay half-time at late times $t \gg t_m$ are given by

$$t_{1/2} = \frac{\tau_{1/2}}{a_0} \approx \frac{0.693}{a_0[J_\infty - (1 - k)]}$$

and

$$t_2 = \frac{\tau_2}{a_0} = \frac{\ln \sqrt{2}}{a_0 c_3} \approx \frac{0.347}{a_0 c_3}$$

(2.32)

whereas the half-early-peak law in real time becomes

$$\frac{\dot{J}_{1/2}}{\dot{J}_{\max}} = \left[ \frac{2(J_{1/2} - J(t_0)) \cosh [c_3 a_0 (t_m - t_0)/2]}{J_0 - J(t_0)} \right]^2.$$

(2.33)

This will be made more precise below, as we apply the semi-time SIR model to two waves, each with its own onset.

# 3. Applications

The above derivations resulted in explicit expressions for the dimensionless fraction $J$ of infected persons and the dimensionless rate of infections $j$, both in terms of reduced time $\tau$, the inverse reproduction number $k$ and the initial condition $J(0) = \eta$. Measured, reliable data is usually available for the total number of deceased persons $D(t)$ as function of time $t$ in units of days, and the total population $N$ can be considered known. Since the number of deaths follows the number of infections with a delay of about 10 days [2], we can use $D(t)$ to make predictions about the number of infected persons at $t - 10$ days.

To uniquely determine the model parameters from the data, which allows us then to draw conclusions about the future time-evolution of measurable quantities, we need to assume a fixed relationship between real and reduced time. To this end, we adopt for the time during each of the pandemic waves a constant infection rate $a_0$ so that $\tau = \int_{t_0}^{t} \mathrm{d}\xi a(\xi) = a_0(t - t_0)$ where $t_0$ is the real time marking the beginning of the $n$th wave, and $\tau$ the reduced time that vanishes at the beginning of the $n$th wave.

With the known fatality ratio of $f \approx 0.005$, the cumulative number of infected persons (including those that have not been identified) is $fD(t)/N$. More precisely, $J(\tau) = f\,D(t)/N$ during the first wave. Because the cumulative number accumulates during subsequent waves, it is more convenient to model the measured daily rate $\dot{J}(t)$ of newly infected persons instead of the cumulative numbers. More precisely, one has $j(\tau) = f\dot{D}(t)/a_0 N$, and it is this dimensionless $j(\tau)$ which we have expressed in terms of $k$ and $\eta$ above, while $\eta$ is contained in the initial condition, $j(0) = \eta(1 - \eta) = f\dot{D}(t_0)/a_0 N$. For each wave, there are thus three parameters to be determined, $k$, $\eta$ and $a_0$, or alternatively, $k$, $t_0$ and $a_0$. In practice, as the measured data is fluctuating considerably, especially at the beginning of a pandemic wave, it turns out to be even more convenient to work with four unknowns, $k$, $t_0$, $\eta$, and $a_0$, and to determine these parameters upon requiring that the absolute deviation between measured and predicted $j(\tau)$ achieves a global minimum for each wave separately. Start values for the global minimization are readily available from our above consideration about the early time evolution, the position of the maximum in real time, $t_m = t_0 + a_0 \tau_m$, the value of the dimensionless $J(\tau_m) = J_0 = fD(t_0 + \tau_m/a_0)/N$, where $\tau_m$ and $J_0$ are known in terms of $k$ and $\eta$. With the fitted parameters at hand the model-predicted cumulative fraction of infected persons in real time $t$ is $J(t) = \int_{t_0}^{t} \dot{J}(\xi) \mathrm{d}\xi = a_0 \int_{t_0}^{t} j[a_0(t' - t_0)] \mathrm{d}t' = \int_0^\tau j(\tau') \mathrm{d}\tau'$ and the model-predicted number of deceased persons is thus $D(t) = J(t)fN$.

For the sake of clarity, because the number of infections is $f^{-1}$ times the number of fatalities, and because the number of fatalities is more reliably reported than the true number of infections, we do not add any delay time when presenting figures, so that the time of outbreak of the pandemic can be considered 10 days earlier than $t_0$. Similarly, the peak time of daily new infections is roughly 10 days earlier than the tabulated $t_m$, which is the peak time of daily fatalities.

In the following, we determine the two sets of four SIR parameters by fitting the early time evolution before the peaks of first and second waves with the monitored early daily case rates $j(t)$ during the waves. The obtained numbers are tabulated in table 1 for selected countries, while the procedure has been applied to more than 100 countries during the course of this study (supplementary website). The measured data are compared with the SIR predictions in figures 1 and 2. With the four coefficients $c_1$, $c_2$, $t_0$ and $a_0$ determined we then can infer

(1) the values of the parameters $\eta$ and $k$ characterizing the waves,
(2) the final number of infected persons $NJ_\infty$, the maximum daily rate $j_{max}$ at the peak time $\tau_m$ according to equations (2.3), (2.6) and (2.17) and
(3) the late time evolution of the second wave after its peak from equations (2.19) and (2.31) determined by the coefficients $d_1$ and $d_2$ given uniquely by equations (2.11) and (2.14) in terms of $k$, $J_\infty$, $c_0$, $c_1$ and $c_2$, which in turn are given by equations (2.9), (2.10) and (2.13), i.e. finally, in terms of $k$, the initial condition $\varepsilon$, and reduced time $\tau$.

All these results are collected in table 1. In the last column of this table, we have added data that became available in the meantime. Deviations between the early predictions from 11 January 2021 and the reported numbers for the second wave are about 10%. We regard this as a convincing proof that the homogeneous SIR model equations successfully and highly accurately (within 10%) forecast the evolution and outcome of Covid-19 pandemic events. At the same time, the model is not able to capture the onset of a $(n+1)$th wave, if it occurs simultaneously with the $n$th wave. This is evident from the data for Italy shown in figure 1. At the time of submission of the original manuscript, the $\alpha$ variant started to rise in Italy and shortly afterwards gave rise to a third wave. This situation is not captured by the current approach, as it requires a clear time separation between waves. Still, we estimated the second wave to terminate when the new variant started to dominate (figure 3).

Despite the differences in the response of authorities to the emerging pandemic, length and completeness of lockdowns etc. there are very comparable patterns for all countries. Both the inverse reproduction number $k$ and the infection rates $a_0$ have significantly dropped during the second wave when compared with the first wave. The decreasing $k = \mu_0/a_0$ tends to increase the peak height, while the decreasing $a_0$ tends to lower it. The reduction of $a_0$ comes as a surprise to us, but a decrease of $a_0$ is in agreement with the observed broadening of the second wave.

One likely explanation may be that the early phases of the second waves in all considered countries occurred under light lockdown conditions whereas no lockdown measures have been taken during the initial phases of the first wave. Moreover, the infection rates $a_0$ of the second wave have dropped as a result of the overall, increasingly cautious, self-protective behaviour of the population.

While the second peak times are comparable in Italy, Switzerland, France and Russia, they are delayed in Canada, Germany and Great Britain. On the positive side, according to our analysis, the peak time $t_m$ of the second wave has passed already in France, Belgium, Italy, Germany, Switzerland and Russia, and more than half of the population in Great Britain will have been infected already after the second wave, thus getting closer to herd immunity. In general, the second wave increased the immunity more significantly than the first wave, as is obvious from the sharp rise of $J_\infty$ between first and second wave (table 1). The last column in this table contains the SIR predictions for the number of fatalities at the end of the second wave. While countries like Germany were hit by a moderately low number of fatalities during the first wave, the number of fatalities will be increasing by a factor of about 7 during the second wave, despite rigorous interventions. Still, the fraction of seriously affected population remains slightly below in this country compared with countries like Switzerland, that did not install a similarly rigorous scenario with hard lockdowns. The measured data is captured by the semi-time SIR model by a relative error of about 5–10%. At present, the situation in Great Britain is obviously the darkest for these sets of data. The trend is most likely caused by the recent, more transmittable variant (SARS-CoV-2 VUI 202012/01) of the virus. As of today (13 January 2021) the peak time in the death rate in Great Britain is still ahead.

Within the semi-time SIR model, the exponential behaviour of the differential rate of newly infected persons $j(\tau)$ is characterized by the early doubling time $\tau_2$ and the late half decay times $\tau_{1/2}$, which we have included in table 1. Their analytic expressions were given by equations (2.25) and (2.24). It seems that $\tau_{1/2}$ is for all cases reported in our table approximately equal to the doubling time $\tau_2$. To see if this is a generic feature of the SIR model, one has to just inspect the ratio $\tau_2/\tau_{1/2} = (J_\infty - 1 + k)/2c_3$,

**Table 1.** Results from the analysis of $n$th (first and second) pandemic waves in selected countries based on data from 11 January 2021. The columns contain the population $N$, SIR parameters $k$, $\eta$, infection rate $a_0$, time of outbreak $t_0$, the resulting peak time $t_m$, fraction $J_\infty$ of cumulated infected persons at the end of the $n$th wave, the precision (prec.) to which the data was described by the semi-time SIR model, the total number $J_\infty N$ of infected persons at the end of the $n$th wave, the early doubling $\tau_2$ and late half decay times $\tau_{1/2}$ given by equations (2.25) and (2.24), and the total number of projected fatalities $D_\infty$ at the end of the $n$th wave. All the polynomial coefficients $c_0$, $c_1$, $c_2$, $d_1$, $d_2$ and $j_{max}$ characterizing the relationship between daily new fraction, $j$, and cumulative fraction of infected persons, $J$, are expressed in terms of $\eta$ and $k$ within this manuscript. The last column contains total fatalities $D_\infty^{meas}$ reported at the end of the second pandemic wave, beginning of April 2021 for all countries but Italy, whose second wave terminated five weeks earlier, during the rise of a third wave. Numbers in Great Britain are about 10% smaller than predicted, certainly due to a relatively rigorous vaccination campaign.

| country | $\alpha_3$ | wave | $N/10^6$ | $k$ | $\eta N$ | $a_0$ | $t_0$ | $t_m$ | $J_\infty$ | prec. | $J_\infty N$ | $t_2$ | $t_{1/2}$ | $D_\infty$ | $D_\infty^{meas}$ |
|---|---|---|---|---|---|---|---|---|---|---|---|---|---|---|---|
| Italy | ITA | #1 | 60.6 | 0.942 | 294 | 1.49 | 2020-01-11 | 2020-04-03 | 11.4% | 7.3% | 6 892 608 | 8 | 8 | 34 463 | |
| Germany | DEU | #1 | 82.7 | 0.990 | 705 | 9.94 | 2020-03-16 | 2020-04-16 | 2.0% | 7.0% | 1 644 557 | 6 | 6 | 8223 | |
| Switzerland | CHE | #1 | 8.4 | 0.978 | 283 | 5.12 | 2020-03-10 | 2020-04-08 | 4.6% | 8.5% | 385 200 | 6 | 6 | 1926 | |
| Great Britain | GBR | #1 | 65.6 | 0.939 | 306 | 1.52 | 2020-01-29 | 2020-04-17 | 12.0% | 7.0% | 7 901 797 | 7 | 8 | 39 509 | |
| France | FRA | #1 | 66.9 | 0.958 | 501 | 3.17 | 2020-02-23 | 2020-04-09 | 8.3% | 10.1% | 5 546 490 | 5 | 5 | 27 732 | |
| Belgium | BEL | #1 | 11.3 | 0.917 | 897 | 1.62 | 2020-03-07 | 2020-04-13 | 16.2% | 9.4% | 1 840 827 | 5 | 5 | 9204 | |
| Canada | CAN | #1 | 36.3 | 0.976 | 579 | 2.90 | 2020-03-06 | 2020-05-05 | 4.9% | 6.1% | 1 774 863 | 5 | 10 | 8874 | |
| Russia | RUS | #1 | 144.3 | 0.987 | 891 | 2.69 | 2020-03-05 | 2020-06-25 | 2.6% | 9.3% | 3 753 373 | 19 | 20 | 18 767 | |
| Italy | ITA | #2 | 60.6 | 0.915 | 1095 | 0.71 | 2020-08-16 | 2020-12-04 | 27.8% | 6.5% | 16 870 360 | 12 | 13 | 84 352 | 95 600 |
| Germany | DEU | #2 | 82.7 | 0.932 | 2129 | 0.75 | 2020-09-11 | 2021-01-02 | 15.4% | 7.8% | 12 712 430 | 15 | 16 | 63 562 | 74 060 |
| Switzerland | CHE | #2 | 8.4 | 0.906 | 121 | 0.61 | 2020-08-03 | 2020-12-04 | 22.7% | 5.7% | 1 904 648 | 7 | 13 | 9523 | 10 322 |
| Great Britain | GBR | #2 | 65.6 | 0.839 | 1325 | 0.21 | 2020-06-04 | 2021-01-15 | 42.5% | 15.0% | 27 884 238 | 20 | 23 | 139 421 | 127 078 |
| France | FRA | #2 | 66.9 | 0.932 | 4693 | 0.68 | 2020-08-16 | 2020-11-27 | 21.7% | 6.9% | 14 532 325 | 15 | 15 | 72 662 | 76 000 |
| Belgium | BEL | #2 | 11.3 | 0.904 | 2360 | 0.70 | 2020-09-12 | 2020-11-16 | 34.9% | 5.6% | 3 965 235 | 10 | 11 | 19 826 | 21 247 |
| Canada | CAN | #2 | 36.3 | 0.954 | 2345 | 0.69 | 2020-09-07 | 2021-01-13 | 14.1% | 5.1% | 5 131 542 | 21 | 22 | 25 658 | 22 892 |
| Russia | RUS | #2 | 144.3 | 0.951 | 20 158 | 0.58 | 2020-08-23 | 2020-12-20 | 12.4% | 5.7% | 17 892 267 | 23 | 24 | 89 461 | 99 400 |

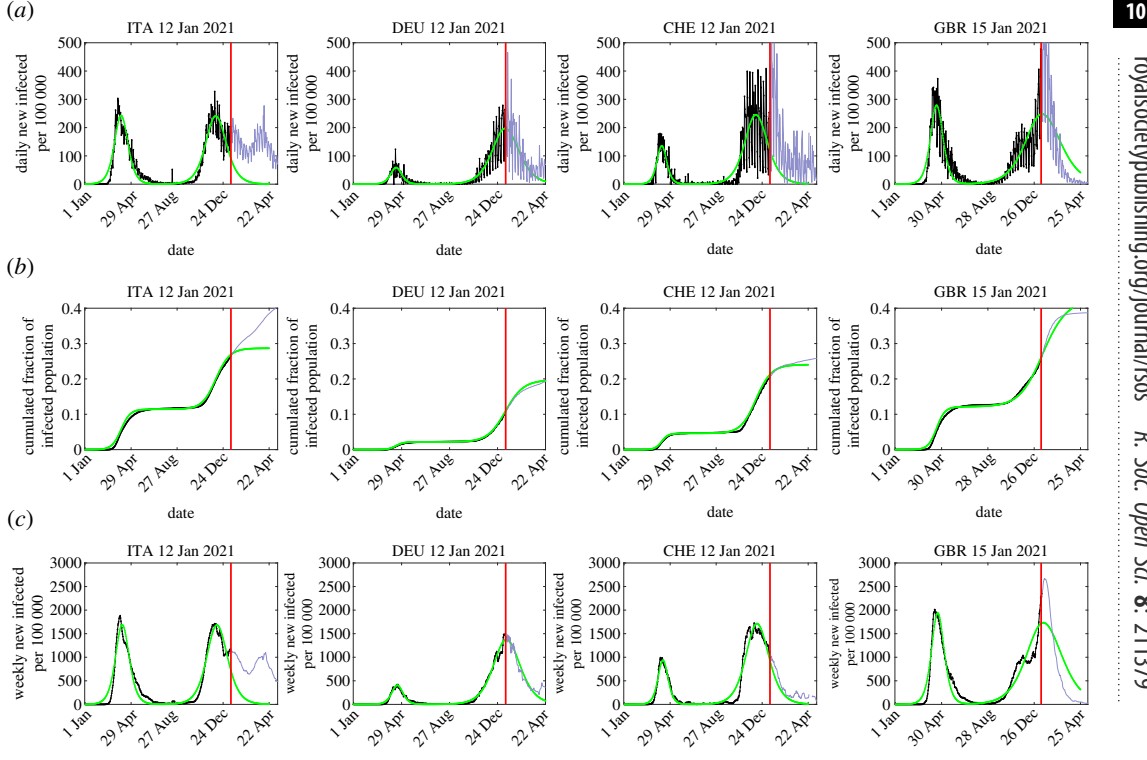

**Figure 1.** Number of infected persons, estimated from the reported fatality data for four countries (black), assuming a fatality rate of $f = 0.005$, along with predictions resulting from the semi-time SIR model (green). The SIR parameters and related quantities are listed in table 1. Shown are both the reported data known at the time of original submission, as well as data collected afterwards. The datasets are separated by a red vertical line. (*a*) Daily new infected persons per 100 000 inhabitants (black), together with the SIR prediction for the second wave (green). (*b*) Cumulated fraction of infected persons, and (*c*) weekly new infected persons per 100 000 inhabitants. Shown here is the analysis of data for Italy, Germany, Switzerland and Great Britain. For additional countries, see figure 2.

using the above explicit expressions for $j_{max}$ and $c_3$. For $k$ close to unity, and small $\eta \ll 1$, the ratio evaluates to

$$\lim_{\eta \to 0} \frac{\tau_2}{\tau_{1/2}} = \frac{1 + 2k}{3} \tag{3.1}$$

in agreement with table (1), as $k$ is sufficiently close to unity. To stress an important aspect, and because we can take the opportunity to compare with related recent work [35], the ratio over the whole $k$ range is shown in figure 4 for various finite $\eta$. This ratio reflects the asymmetry of the pandemic wave, as $J_0 - J_\infty/2$ does. It is important to notice here that the initial condition affects not only the asymmetry of the wave but the asymptotic behaviour qualitatively. The corresponding result for the all-time SIR model is shown for comparison. It is however recovered by the semi-time SIR for sufficiently small $\eta$. The appearance and relevance of power law and Gaussian rather than exponential tails, that are not predicted by the SIR model with time-independent $k$, had been discussed in several recent works [1–4,40]. Early outbreak estimates of $k$ and their uncertainty have been discussed in detail [41].

## 4. Summary and conclusion

We have derived simple analytic expressions for all measurable amounts of cases and fatalities during a pandemic evolution described by the semi-time SIR model, that share all relevant features with the exact solution of the semi-time SIR model, including time and position of the peak of daily new infections, as well as the asymptotic behaviours at small and large times. We show, in particular, how the asymmetry of the epidemic wave and its exponential tails are affected by the initial conditions; a feature that has no analogue in the all-time SIR model. The expressions are so precise that they can be used instead of a

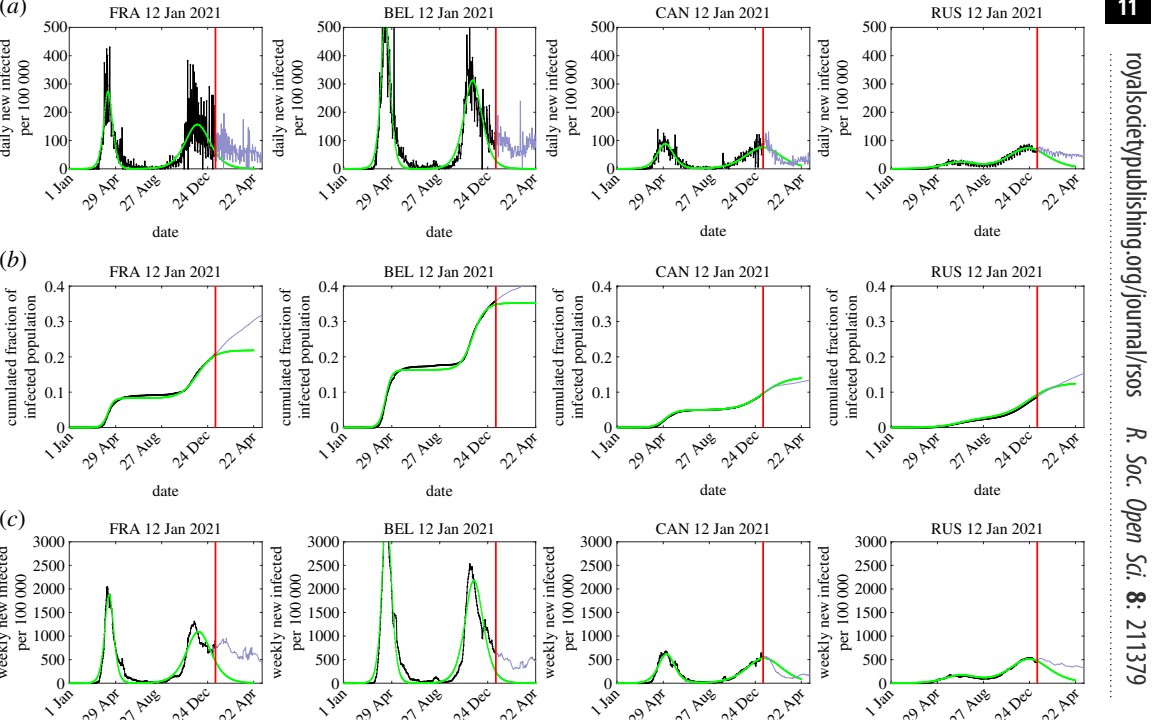

**Figure 2.** Same as figure 1 for additional countries: France, Belgium, Canada and Russia.

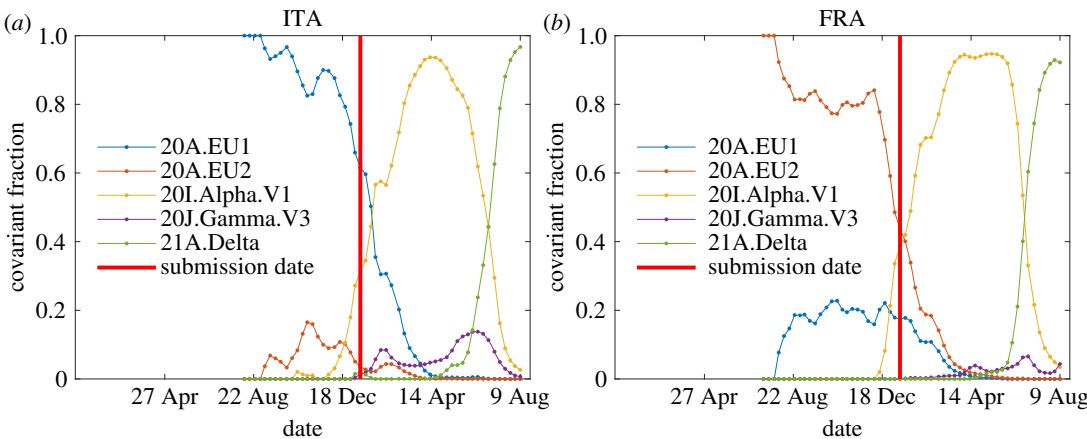

**Figure 3.** Fraction of sequences (not cases) that fall into the variant groups 20A.EU1, 20A.EU2, 20I.Alpha.V1, 20H.Beta.V2, 20J.Gamma.V3 and 21A.Delta [39]. Note that all data are not necessarily representative. Sometimes some samples are more likely to be sequenced than others [39]. Data shown for (a) ITA and (b) FRA (time frame: 1 January 2020 to 9 August 2021). The vertical red line marks the date (16 January 2021) at which this manuscript had originally been submitted. This date happens to roughly coincide with the time of rising relevance of the $\alpha$ variant, while the forecast for the second wave done in the present manuscript assumes an unaltered majority of 20A variants.

numerical solution of the SIR model. The advantage of an analytical expression is obvious, as it allows us to quickly determine the SIR parameters from the measured data well ahead of the peak time, and thus allows for predictions that serve as a prerequisite to make decisions. We applied the approach to second waves in eight different countries from different continents. We summarized the exact features of the semi-time SIR model, stated the approximants for the reliably measurable quantities, and collected all the derivations of the new approximant in an appendix. Our analysis reveals that the immunity is very strongly increasing during the second wave, while it was still at a very moderate level of a few per cent in several countries at the end of the first wave. The wave-specific SIR parameters $\mu_0$ and $a_0$ describing the infection and recovery rates we find to behave in a

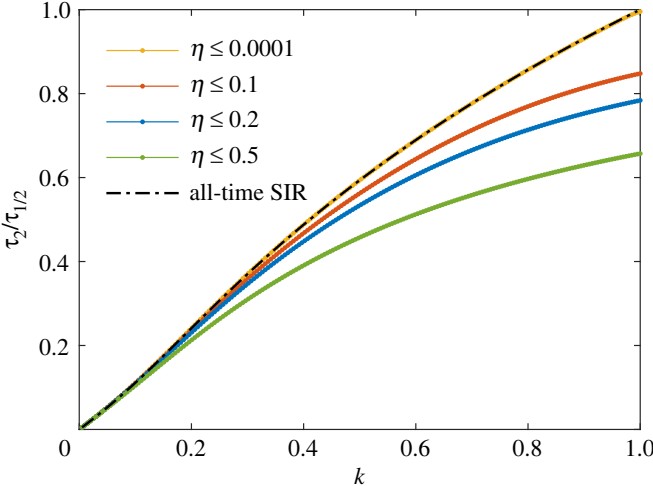

**Figure 4.** Ratio between early doubling time $\tau_2$ and late half decay time $\tau_{1/2}$ versus $k$, for various initial conditions. For comparison, we include the corresponding result [35] for the all-time SIR model, $\tau_2/\tau_{1/2} = k[1 + W_0(-k^{-1}e^{-1/k})]/(1-k)$.

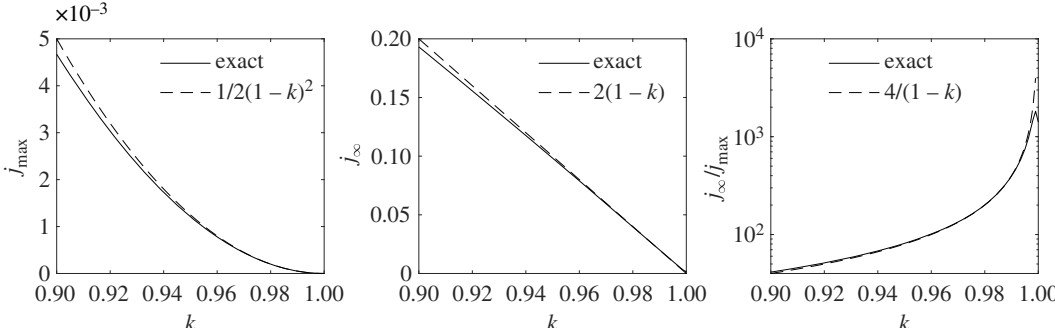

**Figure 5.** (Exact behaviour (solid lines) of the peak differential fraction of infected persons $j_{max}$, the final fraction of infected persons $J_\infty$, and the ratio $J_\infty/j_{max}$, that characterizes the dimensionless width of the wave. All curves are shown to be well captured by the analytic expressions mentioned in the legends (dashed lines), that we use to discuss qualitatively, in concert with the dimensional infection rate $a_0$, the obtained SIR parameters. The exact expressions for $j_{max}$ and $J_\infty$ are given in terms of Lambert's function in equations (2.6) and (2.3). The plot is done with $\eta = 10^{-6}$, but it is essentially unaffected by the choice of $\eta$ as long as $\eta \ll 1$.

similar fashion, while their ratio $k = \mu_0/a_0$ was decreasing only by about 5% for the countries mentioned in table 1 and figures 1 and 2. Still, an apparently moderate change of $k$ can have significant consequences for the relevant numbers like the final number of infected or deceased population, captured by $J_\infty$. For $k$ close to unity and small $\eta \ll 1$, our results imply $j_{max} \approx \frac{1}{2}(1-k)^2$ and $J_\infty \approx 2(1-k)$, cf. figure 5. This implies a typical duration of the differential fraction of newly infected persons, $w \approx J_\infty/a_0 j_{max} \approx 4/a_0(1-k)$, that decreases dramatically with decreasing $k$, but increases with decreasing $a_0$. It is this qualitative feature of the SIR model that leads to its counterintuitive parameters we reveal having analyzed the two pandemic waves, and that has to be taken into account when speculating about possible additional waves. As we have shown, the probability for an additional wave exceeding the peak fatality rate of previous waves is however low in several countries due to the fraction of immune inhabitants at the end of the second wave (second and third for cases where both waves are strongly overlapping), irrespective of the currently ongoing vaccination efforts.

The SIR-modelling is founded on a mean-field approach where the injection and recovery rates are averaged over a large number of persons in the considered countries. Therefore, the well-established influence of pre-symptomatic and asymptomatic disease carriers on the spread of the Covid-19 is washed-out and interwoven with the far more infected persons with less intense pathegonic transmissions. For this reason, our results should only be applied to populations with large enough sizes.

Making use of data that became available during review time allowed us to test the original forecast. The accuracy of our early predictions on the second wave temporal evolution is convincingly good. We predicted the fraction of infected population at the end of the second wave (figure 1). E.g. for ITA we

predicted 0.29 while the true value is 0.33 (13% off), for DEU we predicted 0.20 compared with 0.22 (10% off), for CHE 0.24 versus 0.26 (8% off) and for GBR 0.41 versus 0.40 (2.5% off). This is an outstanding agreement between predictions and measurements given all the parameters of the problem. This comparison proves the high accuracy of the SIR model in predicting the evolution of pandemic outbreaks.

At first sight, it seems that we were unsuccessful in forecasting possible further Covid-19 waves in some countries. However, it is too early for such a statement since so far the appearance of such additional waves did not show up clearly in the very reliable monitored death rates in six out of the eight considered countries (with the exception of Russia and Canada) but only in the monitored rate of new infections which are known to be highly incomplete and therefore less trustworthy. Secondly, our analysis of the second waves has been based on the monitored death rates adopting a mortality rate of 1/200 with respect to the rate of new infections, and it has assumed a clear separation of second and potential third wave in time. The model could be extended to consider the occurrence of simultaneous waves, non-local effects, multiple seeds, spreaders or outbreaks to the expense of additional parameters, with the help of numerical simulation [42–56].

Data accessibility. All data points shown in this contribution are available from https://doi.org/10.5061/dryad.jsxksn09n. The data used to create all graphs that mentioned reported numbers were collected from the following website at github: https://pomber.github.io/covid19/timeseries.json. See https://github.com/pomber/covid19 and https://github.com/CSSEGISandData/COVID-19 and [57] for a description of the data sources. The analysis has been performed for many additional countries. Results are available online at https://www.complexfluids.ethz.ch/covid19-wave2.

Authors' contributions. Both authors contributed equally to this work.

Competing interests. We declare we have no competing interests.

Funding. No funding has been received for this article.

Acknowledgements. M.K. thanks the Swiss National Supercomputing Centre for providing computational resources (CSCS projects go11 and s987).

# Appendix A. Calculation of integrals

Equation (2.18) can be written as

$$\tau - \tau_m \simeq \begin{cases} -I_c & \text{for } \tau \leq \tau_m \\ I_d & \text{for } \tau \geq \tau_m, \end{cases} \tag{A 1}$$

in terms of the three integrals $\tau_m$, $I_c$ and $I_d$,

$$\tau_m = \int_\eta^{J_0} \frac{\mathrm{d}y}{j_c(y)} = \int_\eta^{J_0} \frac{\mathrm{d}y}{c_0 + c_1(y - \eta) + c_2(y - \eta)^2}$$
$$= \int_0^{J_0 - \eta} \frac{\mathrm{d}x}{c_0 + c_1 x + c_2 x^2}, \tag{A 2}$$

$$I_c = \int_J^{J_0} \frac{\mathrm{d}y}{j_c(y)} = \int_{J-\eta}^{J_0 - \eta} \frac{\mathrm{d}x}{c_0 + c_1 x + c_2 x^2} \tag{A 3}$$

$$I_d = \int_{J_0}^{J} \frac{\mathrm{d}y}{j_d(y)} = \int_J^{J_0} \frac{\mathrm{d}y}{d_1(J_\infty - y) + d_2(J_\infty - y)^2}$$
$$= \int_{J_\infty - J_0}^{J_\infty - J} \frac{\mathrm{d}x}{x(d_1 + d_2 x)} = \int_{1/(J_\infty - J_0)}^{1/(J_\infty - J)} \frac{\mathrm{d}w}{d_2 + d_1 w}, \tag{A 4}$$

where in the last step we substituted $x = 1/w$. The integral (A4) then becomes for $d_1 \neq 0$

$$I_d = \frac{1}{d_1} \ln \frac{1 + (d_1/d_2(J_\infty - J))}{1 + (d_1/d_2(J_\infty - J_0))}. \tag{A 5}$$

As the coefficient $c_2 < 0$ is negative the expression

$$2c_3 = \sqrt{c_1^2 - 4c_0 c_2} > 0, \tag{A 6}$$

is always real-valued and positive. Consequently, the two integrals (A 2) and (A 3) are given by

$$
\begin{aligned}
\tau_m &= \frac{1}{2c_3}\left[\ln\frac{c_2 x + ((c_1 - 2c_3)/2)}{c_2 x + ((c_1 + 2c_3)/2)}\right]_0^{J_0 - \eta} \\
&= \frac{1}{2c_3}\left[\ln\frac{1 - (2c_3/(c_1 + 2c_2 x))}{1 + (2c_3/(c_1 + 2c_2 x))}\right]_0^{J_0 - \eta} \\
&= -\frac{1}{c_3}\left[\operatorname{artanh}\frac{2c_3}{c_1 + 2c_2 x}\right]_0^{J_0 - \eta} \\
&= \frac{1}{c_3}\left[\operatorname{artanh}\frac{2c_3}{c_1} - \operatorname{artanh}\frac{2c_3}{c_1 + 2c_2(J_0 - \eta)}\right] \\
&= \frac{1}{c_3}\left[\operatorname{artanh}\frac{4c_2 c_3(J_0 - \eta)}{c_1^2 - 4c_3^2 + 2c_1 c_2(J_0 - \eta)}\right] \\
&= \frac{1}{c_3}\operatorname{artanh}\frac{2c_3}{c_1 + (2c_0/(J_0 - \eta))}
\end{aligned}
\tag{A 7}
$$

and

$$
\begin{aligned}
I_c &= \frac{1}{2c_3}\left[\ln\frac{c_2 x + ((c_1 - 2c_3)/2)}{c_2 x + ((c_1 + 2c_3)/2)}\right]_{J - \eta}^{J_0 - \eta} \\
&= \frac{1}{c_3}\left[\operatorname{artanh}\frac{2c_3}{c_1 + 2c_2 x}\right]_{J_0 - \eta}^{J - \eta} \\
&= \frac{1}{c_3}\operatorname{artanh}\frac{4c_2 c_3(J_0 - J)}{c_1^2 - 4c_3^2 + 2c_1 c_2\beta_1 + 4c_2^2\beta_2} \\
&= \frac{1}{c_3}\operatorname{artanh}\frac{2c_3(J_0 - J)}{2c_0 + c_1\beta_1 + 2c_2\beta_2},
\end{aligned}
\tag{A 8}
$$

where in both calculations we made use of equation (A 6) in the last step. To make the expression (A8) more readable, we used the abbreviations $\beta_1 = J_0 + J - 2\eta$ and $\beta_2 = (J_0 - \eta)(J - \eta)$. As an aside, we note that for $J = \eta$ the last formula (A 8) correctly reduces to equation (A 7).

## A.1. Early reduced times $\tau \leq \tau_m$

Collecting terms in equation (A 1), we obtain for early reduced times $\tau \leq \tau_m$ with the abbreviation $X = c_3(\tau_m - \tau)$ as well as

$$
Y = J - \eta, \quad Y_0 = J_0 - \eta
\tag{A 9}
$$

and

$$
T(\tau) = -\frac{1}{2c_3}\tanh(X),
\tag{A 10}
$$

the linear relation

$$
\begin{aligned}
Y - Y_0 &= [2c_0 + c_1(Y + Y_0) + 2c_2 Y Y_0]T \\
&= [2c_0 + c_1 Y_0 + (c_1 + 2c_2 Y_0)Y]T.
\end{aligned}
\tag{A 11}
$$

Inserting equation (2.13) for $c_2$, we find

$$
c_1 + 2c_2 Y_0 = \frac{2j_{\max} - (2c_0 + c_1 Y_0)}{Y_0},
\tag{A 12}
$$

simplifying equation (A 11) to

$$
Y - Y_0 = \left[(2c_0 + c_1 Y_0)\left(1 - \frac{Y}{Y_0}\right) + \frac{2j_{\max}Y}{Y_0}\right]T.
\tag{A 13}
$$

From equations (A 10) and (A 7), we identify

$$
T(0) = T(\tau = 0) = -\frac{Y_0}{2c_0 + c_1 Y_0},
\tag{A 14}
$$

so that equation (A 13) becomes

$$
\frac{2j_{\max}Y}{Y_0(Y - Y_0)} = \frac{1}{T} - \frac{1}{T(0)},
\tag{A 15}
$$

and equation (A 15) is solved by

$$Y(T) = \frac{Y_0}{1 - (A/((1/T) - (1/T(0))))} = \frac{Y_0}{1 + (AT(0)T/(T - T(0)))}, \tag{A 16}$$

with

$$A = \frac{2j_{\max}}{Y_0}. \tag{A 17}$$

Using equation (A 10) then provides

$$\frac{1}{T} - \frac{1}{T(0)} = -2c_3[\coth(X) + \coth(c_3\tau_m)]$$

$$= -2c_3 \frac{\sinh(c_3\tau)}{\sinh(X)\sinh(c_3\tau_m)}. \tag{A 18}$$

Consequently, equation (A 16) reduces to

$$Y(\tau \le \tau_m) = \frac{Y_0}{1 + (A/2c_3)(\sinh(X)\sinh(c_3\tau_m)/\sinh(c_3\tau))}$$

$$= \frac{Y_0}{1 + (j_{\max}/Y_0c_3)(\sinh(X)\sinh(c_3\tau_m)/\sinh(c_3\tau))}. \tag{A 19}$$

We next offer two different routes A and B to come up with equivalent, but formally very different expressions for both $Y(\tau \le \tau_m)$ and $J(\tau \le \tau_m)$. Route (A): Using $\tau_m$ from equation (A 7), we find

$$\sinh(c_3\tau_m) = \frac{\tanh(c_3\tau_m)}{\sqrt{1 - \tanh^2(c_3\tau_m)}}$$

$$= \frac{c_3Y_0}{\sqrt{c_0^2 + c_0c_1Y_0 + c_0c_2Y_0^2}}$$

$$= \frac{c_3Y_0}{\sqrt{c_0[c_0 + c_1Y_0 + c_2Y_0^2]}}. \tag{A 20}$$

According to the first equation (2.15)

$$j_{\max} = j(J_0) = c_0 + c_1Y_0 + c_2Y_0^2 \tag{A 21}$$

so that equation (A 20) becomes

$$\sinh(c_3\tau_m) = \frac{c_3Y_0}{\sqrt{c_0 j_{\max}}}, \tag{A 22}$$

implying for equation (A 19)

$$Y(\tau \le \tau_m) = \frac{Y_0}{1 + \sqrt{j_{\max}/c_0}(\sinh(X)/\sinh(c_3\tau))} \tag{A 23}$$

and hence

$$J(\tau \le \tau_m) = \eta + \frac{J_0 - \eta}{1 + \sqrt{\dfrac{j_{\max}}{c_0}}\dfrac{\sinh(X)}{\sinh(c_3\tau)}}, \tag{A 24}$$

confirming the first case in equation (2.19). For $\tau = \tau_m$, this solution correctly reduces to $J_0$, whereas for $\tau = 0$ it correctly reduces to $\eta$.

Route (B): To derive alternative expressions, we start from equation (A 19) and note that equation (A 7) provides

$$\tanh(c_3\tau_m) = \frac{2c_3Y_0}{2c_0 + c_1Y_0}, \tag{A 25}$$

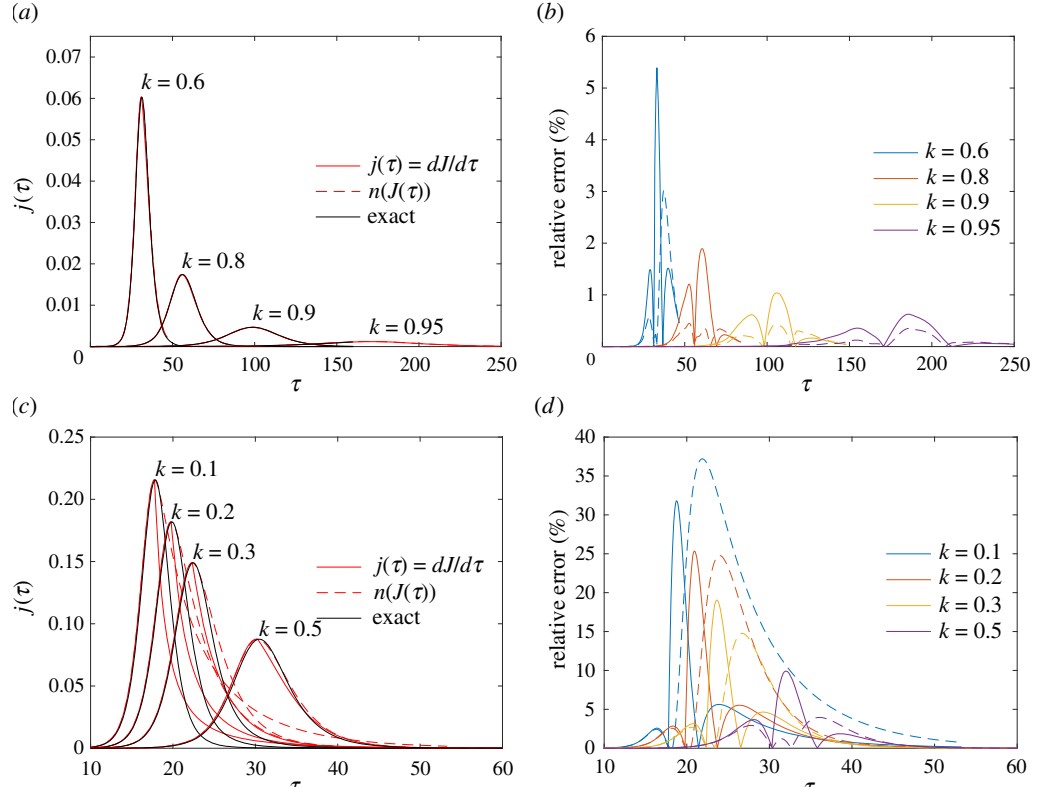

**Figure 6.** (a,c) Comparison between exact and here developed approximate solutions to the semi-time SIR model, for various $k$. (b,d) Relative error of the approximants shown in (a,b): $j(\tau)$ from equation (2.22), which is identical with $dJ/\tau$ and also identical with $j(\tau)$ according to equation (2.15) (solid) and $j(\tau) = n(J(\tau))$ according to equation (2.2), with $J(\tau)$ from equation (2.19). The plot is done with $\eta = 10^{-6}$, but it is qualitatively unaffected by the choice of $\eta$, not only for small $\eta \ll 1$.

so that

$$\left.\begin{array}{r}
\cosh(c_3\tau_m) = \dfrac{1}{\sqrt{1 - \tanh^2(c_3\tau_m)}} = \dfrac{2c_0 + c_1 Y_0}{2\sqrt{c_0\, j_{\max}}} \\[12pt]
\text{and} \qquad \sinh(c_3\tau_m) = \dfrac{\tanh(c_3\tau_m)}{\sqrt{1 - \tanh^2(c_3\tau_m)}} = \dfrac{c_3 Y_0}{\sqrt{c_0\, j_{\max}}}.
\end{array}\right\} \qquad (A\,26)$$

The latter two equations can be combined to yield

$$\frac{j_{\max}}{c_3 Y_0} = \frac{1 + (c_1 Y_0/2c_0)}{\sinh(c_3\tau_m)\cosh(c_3\tau_m)}. \qquad (A\,27)$$

Inserting equation (A27) into equation (A19) leads to the alternative expression for $Y$,

$$\begin{aligned}
Y(\tau \le \tau_m) &= \frac{Y_0}{1 + (1 + (c_1 Y_0/2c_0))(\sinh(c_3(\tau_m - \tau))/\cosh(c_3\tau_m)\sinh(c_3\tau))} \\[10pt]
&= \frac{Y_0}{1 + (1 + (c_1 Y_0/2c_0))[\tanh(c_3\tau_m)\coth(c_3\tau) - 1]} \\[10pt]
&= \frac{Y_0}{(c_3 Y_0/c_0)\coth(c_3\tau) - (c_1 Y_0/2c_0)} = \frac{c_0}{c_3\coth(c_3\tau) - (c_1/2)} \\[10pt]
&= \frac{c_0}{c_3\coth(c_3\tau) - (c_1/2)} \\[10pt]
&= \frac{c_0(1 - e^{-2c_3\tau})}{c_3 - (c_1/2) + (c_3 + (c_1/2))e^{-2c_3\tau}},
\end{aligned} \qquad (A\,28)$$

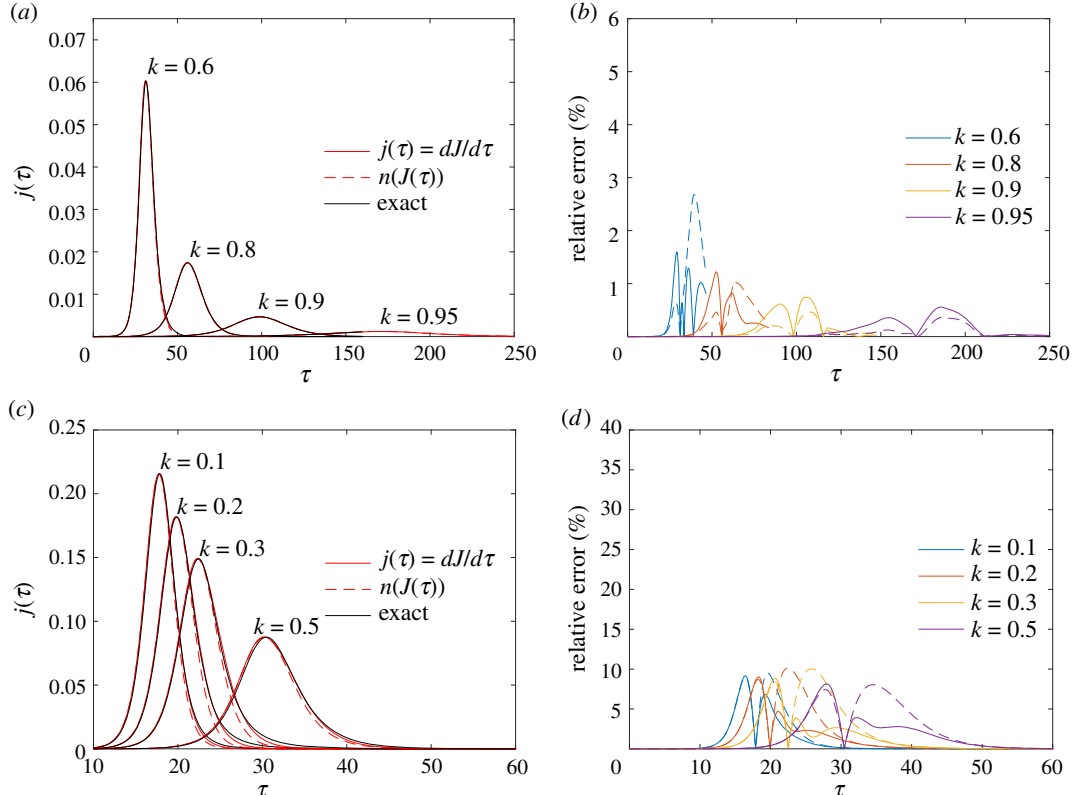

**Figure 7.** (*a,c*) Comparison between exact and approximate solutions to the semi-time SIR model, denoted as version II in our earlier work [36], for various *k*. (*b,d*) Relative error of the approximants shown in (*a,b*): $j(\tau)$ from equation (2.22), which is identical with $dJ/\tau$ and also identical with $j(\tau)$ according to equation (2.15) (solid) and $j(\tau) = n(J(\tau))$ according to equation (2.2), with $J(\tau)$ from equation (2.19). The plot is done with $\eta = 10^{-6}$, but it is qualitatively unaffected by the choice of $\eta$, not only for small $\eta \ll 1$. While for small *k* the error is significantly smaller for this approximant compared to the approximant used in the present work, the present approximant has the neat features that it exactly reaches the exact $j_{max}$ at peak time, and that it possesses the correct asymptotic behaviour, which is not reflected by watching the relative error. Furthermore, the present approximant has a smaller error than the so-called version I approximant [36].

where we used $\sinh[c_3(\tau_m - \tau)] = \sinh(c_3\tau_m)\cosh(c_3\tau) - \cosh(c_3\tau_m)\sinh(c_3\tau)$ and equation (A 25). Consequently,

$$J(\tau \le \tau_m) = \eta + \frac{\eta(1-\eta)}{c_3\coth(c_3\tau) - (c_1/2)}. \tag{A 29}$$

This variant is particularly useful to evaluate the limiting case $k = 0$. Here we have $c_1 = 1 - 2\eta$, $c_2 = -1$ and $c_3 = 1/2$, so that $c_3 - (c_1/2) = \eta$ and $c_3 + (c_1/2) = 1 - \eta$. In this case, equation (A 29) simplifies to

$$J(\tau \le \tau_m) = \frac{\eta}{\eta + (1-\eta)e^{-\tau}} \quad (k = 0). \tag{A 30}$$

Taking the derivative of equation (A 24), our result from route A, we obtain the corresponding rate

$$\begin{aligned} j(\tau \le \tau_m) = \frac{dJ(\tau \le \tau_m)}{d\tau} &= \frac{Y_0 c_3 \sqrt{j_{max}/c_0}}{[1 + \sqrt{j_{max}/c_0}(\sinh(X)/\sinh(c_3\tau))]^2} \\ &\times \frac{\sinh(c_3\tau)\cosh(X) + \cosh(c_3\tau)\sinh(X)}{\sinh^2(c_3\tau)} \\ &= \frac{Y_0 c_3 \sqrt{j_{max}/c_0}}{[1 + \sqrt{j_{max}/c_0}(\sinh(X)/\sinh(c_3\tau))]^2} \frac{\sinh(c_3\tau_m)}{\sinh^2(c_3\tau)} \\ &= j_{max}\left(\frac{\sinh(c_3\tau_m)}{\sinh(c_3\tau) + \sqrt{j_{max}/c_0}\sinh(X)}\right)^2, \end{aligned} \tag{A 31}$$

where we used equation (A 22) and the earlier introduced abbreviation $X = c_3(\tau_m - \tau)$. The same result is obtained if we insert $J(\tau)$ from equation (2.19) into (2.15). This expression (A 31) confirms the first case in equation (2.22). Using instead equation (A 29) obtained in route B, equation (A 31) can alternatively be written as

$$j(\tau \leq \tau_m) = \frac{4c_0 c_3^2}{[c_1 \sinh(c_3 \tau) - 2c_3 \cosh(c_3 \tau)]^2} \tag{A 32}$$

## A.2. Late reduced times $\tau \geq \tau_m$

Likewise, for large reduced times $\tau \geq \tau_m$ we obtain for equation (2.1)

$$\left[1 + \frac{d_1}{d_2(J_\infty - J_0)}\right] e^{d_1(\tau - \tau_m)} - 1 = \frac{d_1}{d_2(J_\infty - J)}, \tag{A 33}$$

yielding

$$J(\tau \geq \tau_m) = J_\infty - \frac{J_\infty - J_0}{(1 + (d_2(J_\infty - J_0)/d_1)) \, e^{d_1(\tau - \tau_m)} - (d_2(J_\infty - J_0)/d_1)}. \tag{A 34}$$

From equation (2.14), we find

$$\frac{d_2(J_\infty - J_0)}{d_1} = \frac{j_{max}}{d_1(J_\infty - J_0)} - 1, \tag{A 35}$$

so that

$$J(\tau \geq \tau_m) = J_\infty - \frac{J_\infty - J_0}{(j_{max}/d_1(J_\infty - J_0))[e^{d_1(\tau - \tau_m)} - 1] + 1}, \tag{A 36}$$

in agreement with the second case in equation (2.19). The late time solution correctly provides $J_0$ for $\tau = \tau_m$ and $J_\infty$ in the limit $\tau \to \infty$.

Taking the derivative of equation (A 36) with respect to $\tau$, the corresponding rate becomes

$$j(\tau \geq \tau_m) = \frac{j_{max}^{d_1(\tau - \tau_m)} e}{\{1 + (j_{max}/d_1(J_\infty - J_0))[e^{d_1(\tau - \tau_m)} - 1]\}^2}, \tag{A 37}$$

confirming the second case in equation (2.22).

# Appendix B. Dynamics in real time

As data are usually available in real time $t$, we here write down the cumulative and differential rates of newly infected persons, which follow from their dimensionless counterparts equations (2.19) and (2.22), upon using the transformation (2.31). The cumulative number of infected persons at time $t$ is

$$J(t) \simeq \begin{cases} \eta + \frac{J_0 - \eta}{1 + \sqrt{j_{max}/c_0}(\sinh[c_3 a_0(t_m - t)]/\sinh(c_3 a_0(t - t_0)))} & \text{for } t \leq t_m \\ J_\infty - \frac{J_\infty - J_0}{(j_{max}/d_1(J_\infty - J_0))[e^{d_1 a_0(t - t_m)} - 1] + 1} & \text{for } t \geq t_m \end{cases} \tag{B 1}$$

and the differential rate $\dot{J}(t) = dJ(t)/dt$ reads

$$\frac{\dot{J}(t)}{a_0 j_{max}} = \begin{cases} \left[\frac{\sinh[c_3 a_0(t_m - t_0)]}{\sinh[c_3 a_0(t - t_0)] + \sqrt{j_{max}/c_0} \sinh[c_3 a_0(t_m - t)]}\right]^2 & \text{for } t \leq t_m \\ \frac{e^{d_1 a_0(t - t_m)}}{(1 + (j_{max}/d_1(J_\infty - J_0))[e^{d_1 a_0(t - t_m)} - 1])^2} & \text{for } t \geq t_m \end{cases} \tag{B 2}$$

where all coefficients $c_0, c_3, d_1, J_0, J_\infty, j_{max}$ are given in terms of $k$ and $\eta = J(0)$ in the above equations (2.9), (2.21), (2.11), (2.4), (2.3) and (2.6), respectively.

# Appendix C. Quality of the approximant

Here, we evaluate the quality of the analytic approximant for the solution $j(\tau)$ of the semi-time SIR model. These calculations can be done for all remaining SIR quantities $S(\tau)$, $I(\tau)$, $R(\tau)$ as well as $J(\tau)$, but the results

are very comparable. In figure 6*a*, we show the reduced time evolution of $j(\tau)$ for various $k$. The exact solution (solid black) is visually matching the approximant (2.22) (solid red). For comparison, we include (dashed red) the $j(\tau)$ calculated via the identity (2.2). The relative error of the approximant, defined by the difference between approximant and exact solution, divided by $j_{\max}$, is shown in figure 6*b*, for the same $k$'s, and again for our approximant (solid) and for the $j(\tau)$ calculated from the approximant $J(\tau)$ given by equation (2.19), and inserted into equation (2.2). The relative error of the approximant is below 2% for the relevant regime of $k \geq 8$, and drops to about 5% for $k = 0.6$. It is important to realize from figure 6*b* that the error is only large during a small interval in time, but afterwards vanishes again, so that the deviations are negligible for any practical purposes. Similarly, deviations between exact and analytic approximant for $J(\tau)$ vanish exactly at $\tau = \tau_m$ and for both small $\tau \ll \tau_m$, and large times $\tau \gg \tau_m$. This means for the differential rate $j(\tau)$ that deviations are not accumulating, but compensating in time. Since the measured data is also fluctuating by amounts that easily exceed 10% between subsequent days, the precision can be considered excellent. For the readers' convenience, we include a similar comparison for the so-called version II approximant we derived in our previous work (figure 7). The present approximant has the neat features that it exactly reaches the exact $j_{\max}$ at peak time $\tau_m$, and that it possesses the correct asymptotic behaviour, which is not fully reflected by watching the relative error. Furthermore, the present approximant has a smaller error than the so-called version I approximant [36], and a simpler analytic form than the so-called version II approximant [36].

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
