## [Peer Review File · Royal Society Open Science]

Review History

RSOS-210595.R0 (Original submission)

Review form: Reviewer 1

Is the manuscript scientifically sound in its present form?

Yes

Are the interpretations and conclusions justified by the results?

Yes

Is the language acceptable?

Yes

Do you have any ethical concerns with this paper?

No

Have you any concerns about statistical analyses in this paper?

No

Recommendation?

Major revision is needed (please make suggestions in comments)

Comments to the Author(s)

The authors have revised the manuscript, which has improved the quality of the manuscript. However, it is still not topical and I am not sure how it might contribute to the fight against the ongoing pandemic, especially since it does not address the impact of non pharmaceutical interventions and vaccines on predictions of subsequent waves.

I am not satisfied with the justification for not including the pre-symptomatic and asymptomatic classes, which both contribute to diseases spread, especially when data is considered. The data considered is only for the confirmed cases, but it is not clear how the authors isolate only these confirmed cases from their model to associate with the data.

The initial conditions used to produce some of the figures, e.g., Figures 1 and 2 are not provided meaning it difficult to reproduce the figures.

The paper is still too technical and more suitable for a more specialized methods or mathematical journal than for a general audience.

Finally, there are minor typos worth correcting in the manuscript. For example, the authors say "amount of cases" instead of "number of cases" in a number of places including the abstract. They also start the names of months with small letters, e.g., april 2021 in the legend of Table 1 and 11 jan 2021 in line 3 on page 18.

Decision letter (RSOS-210595.R0)

Dear Dr Kroger

The Editors assigned to your paper RSOS-210595 "Forecast made on January 11, 2021 for the second Covid-19 wave based on the improved SIR model with a constant ratio of recovery to infection rate" have now received comments from reviewers and would like you to revise the paper in accordance with the reviewer comments and any comments from the Editors. Please note this decision does not guarantee eventual acceptance.

Please submit your revised manuscript and required files (see below) no later than 21 days from today's (ie 05-Aug-2021) date. Note: the ScholarOne system will 'lock' if submission of the revision is attempted 21 or more days after the deadline. If you do not think you will be able to meet this deadline please contact the editorial office immediately.

on behalf of Professor Tim Rogers (Associate Editor) and Mark Chaplain (Subject Editor)
openscience@royalsociety.org

Reviewer comments to Author:

Reviewer: 1

Comments to the Author(s)

The authors have revised the manuscript, which has improved the quality of the manuscript. However, it is still not topical and I am not sure how it might contribute to the fight against the ongoing pandemic, especially since it does not address the impact of non pharmaceutical interventions and vaccines on predictions of subsequent waves.

I am not satisfied with the justification for not including the pre-symptomatic and asymptomatic classes, which both contribute to diseases spread, especially when data is considered. The data considered is only for the confirmed cases, but it is not clear how the authors isolate only these confirmed cases from their model to associate with the data.

The initial conditions used to produce some of the figures, e.g., Figures 1 and 2 are not provided meaning it difficult to reproduce the figures.

The paper is still too technical and more suitable for a more specialized methods or mathematical journal than for a general audience.

Finally, there are minor typos worth correcting in the manuscript. For example, the authors say "amount of cases" instead of "number of cases" in a number of places including the abstract. They also start the names of months with small letters, e.g., april 2021 in the legend of Table 1 and 11 jan 2021 in line 3 on page 18.

===PREPARING YOUR MANUSCRIPT===

one version identifying all the changes that have been made (for instance, in coloured highlight, in bold text, or tracked changes);
 a 'clean' version of the new manuscript that incorporates the changes made, but does not highlight them. This version will be used for typesetting if your manuscript is accepted.

===PREPARING YOUR REVISION IN SCHOLARONE===

- Any electronic supplementary material (ESM).
- If you are requesting a discretionary waiver for the article processing charge, the waiver form must be included at this step.
- If you are providing image files for potential cover images, please upload these at this step, and inform the editorial office you have done so. You must hold the copyright to any image provided.
- A copy of your point-by-point response to referees and Editors. This will expedite the preparation of your proof.

- Ensure that your data access statement meets the requirements at <https://royalsociety.org/journals/authors/author-guidelines/#data>. You should ensure that you cite the dataset in your reference list. If you have deposited data etc in the Dryad repository, please include both the 'For publication' link and 'For review' link at this stage.
- If you are requesting an article processing charge waiver, you must select the relevant waiver option (if requesting a discretionary waiver, the form should have been uploaded at Step 3 'File upload' above).
- If you have uploaded ESM files, please ensure you follow the guidance at <https://royalsociety.org/journals/authors/author-guidelines/#supplementary-material> to include a suitable title and informative caption. An example of appropriate titling and captioning may be found at https://figshare.com/articles/Table_S2_from_Is_there_a_trade-off_between_peak_performance_and_performance_breadth_across_temperatures_for_aerobic_scope_in_teleost_fishes_/3843624.

Author's Response to Decision Letter for (RSOS-210595.R0)

See Appendix A.

Decision letter (RSOS-210595.R1)

Dear Dr Kroger

The Editors assigned to your paper RSOS-210595.R1 "Forecast made on January 11, 2021 for the second Covid-19 wave based on the improved SIR model with a constant ratio of recovery to infection rate" have made a decision based on their reading of the paper and any comments received from reviewers.

Regrettably, in view of the reports received, the manuscript has been rejected in its current form. However, a new manuscript may be submitted which takes into consideration these comments.

We invite you to respond to the comments supplied below and prepare a resubmission of your manuscript. Below the referees' and Editors' comments (where applicable) we provide additional requirements. We provide guidance below to help you prepare your revision.

Please note that resubmitting your manuscript does not guarantee eventual acceptance, and we do not generally allow multiple rounds of revision and resubmission, so we urge you to make every effort to fully address all of the comments at this stage. If deemed necessary by the Editors, your manuscript will be sent back to one or more of the original reviewers for assessment. If the original reviewers are not available, we may invite new reviewers.

Please resubmit your revised manuscript and required files (see below) no later than 14-Feb-2022. Note: the ScholarOne system will 'lock' if resubmission is attempted on or after this deadline. If you do not think you will be able to meet this deadline, please contact the editorial office immediately.

Please note article processing charges apply to papers accepted for publication in Royal Society Open Science (<https://royalsocietypublishing.org/rsos/charges>). Charges will also apply to papers transferred to the journal from other Royal Society Publishing journals, as well as papers submitted as part of our collaboration with the Royal Society of Chemistry (<https://royalsocietypublishing.org/rsos/chemistry>). Fee waivers are available but must be requested when you submit your manuscript (<https://royalsocietypublishing.org/rsos/waivers>).

Thank you for submitting your manuscript to Royal Society Open Science and we look forward to receiving your resubmission. If you have any questions at all, please do not hesitate to get in touch.

on behalf of Professor Tim Rogers (Associate Editor) and Mark Chaplain (Subject Editor)
openscience@royalsociety.org

Associate Editor Comments to Author (Professor Tim Rogers):

Associate Editor

Comments to the Author:

I am satisfied that the authors have addressed the criticisms of the referee, but I do not think we can publish the paper.

It is long past the point where a second wave forecast made on 11 January could be considered timely, since it is now 16 August and many places around the world are well into their third wave. Part of this delay is of course due to the refereeing process, but this is unavoidable.

It is not appropriate to publish an out of date prediction without comparing it to the reality which has now unfolded. A quick look at Figure 1 compared to more recent data of cases since January shows the predictions made were not accurate. I can see a route to publication for an article exploring the application of the modelling methodology and unpicking why it failed in this instance. I do not think we can publish the paper as it stands.

===PREPARING YOUR MANUSCRIPT===

===PREPARING YOUR REVISION IN SCHOLARONE===

<https://royalsociety.org/journals/authors/author-guidelines/#supplementary-material> to include a suitable title and informative caption. An example of appropriate titling and captioning may be found at https://figshare.com/articles/Table_S2_from_Is_there_a_trade-off_between_peak_performance_and_performance_breadth_across_temperatures_for_aerobic_sc_ope_in_teleost_fishes_/3843624.

Author's Response to Decision Letter for (RSOS-210595.R1)

See Appendix B.

Decision letter (RSOS-211379.R0)

Dear Dr Kroger,

I am pleased to inform you that your manuscript entitled "Forecast made on January 11, 2021 for the second Covid-19 wave based on the improved SIR model with a constant ratio of recovery to infection rate" is now accepted for publication in Royal Society Open Science.

COVID-19 rapid publication process:

We are taking steps to expedite the publication of research relevant to the pandemic. If you wish, you can opt to have your paper published as soon as it is ready, rather than waiting for it to be published the scheduled Wednesday.

This means your paper will not be included in the weekly media round-up which the Society sends to journalists ahead of publication. However, it will still appear in the COVID-19 Publishing Collection which journalists will be directed to each week (<https://royalsocietypublishing.org/topic/special-collections/novel-coronavirus-outbreak>).

If you wish to have your paper considered for immediate publication, or to discuss further, please notify openscience_proofs@royalsociety.org and press@royalsociety.org when you respond to this email.

on behalf of Professor Tim Rogers (Associate Editor) and Mark Chaplain (Subject Editor)
openscience@royalsociety.org

Appendix A

Eidgenössische Technische Hochschule Zürich
Swiss Federal Institute of Technology Zurich

Martin Kröger, Prof Dr

Polymer Physics, HCP F 48.2
Leopold-Ruzicka-Weg 4
ETH Zürich
CH-8093 Zürich

Phone +41 44 632 6622
Fax +41 44 632 1076
mk@mat.ethz.ch
www.complexfluids.ethz.ch

Editors

09/08/2021 Zurich

Dear Tim, dear Mark, dear Editor,

Resubmission manuscript RSOS-210595

We thank you for having provided very helpful feedback. We have carefully addressed all recommendations made by the referee, and approved the manuscript further (highlighted changes). Unfortunately, the reviewer 1 missed two points: (i) Our table I contains all parameters and initial conditions to reproduce the figures, and (ii) our analysis is based on the number of reported fatalities, not the number of reported infections. We have stated both facts more clearly in the revised manuscript. We have renamed quantity $N(y)$ to $n(y)$ to avoid confusion with the existing population size N . We are positive that the manuscript is of interest for readers of Royal Society Open Science, especially, as the Editors of Journal of the Royal Society Interface asked us to transfer it to your journal, having screened the content. We hope the carefully prepared manuscript is suitable for inclusion into Royal Society Open Science in its present, revised, form. Please find a point-by-point response to reviewers below.

Kind regards,

Response to Reviewer 1

Reviewer: The authors have revised the manuscript, which has improved the quality of the manuscript. However, it is still not topical and I am not sure how it might contribute to the fight against the ongoing pandemic, especially since it does not address the impact of non-pharmaceutical interventions and vaccines on predictions of subsequent waves.

Authors: There seems to be a misunderstanding by the referee which we have caused by not appropriately describing our approach. We have noted now in the revised text after Eq. (1) that with the introduction of the reduced time in Eq. (1) all our analysis includes and addresses the effects of non-pharmaceutical interventions (NPIs). These affect the infection rate $\alpha(t)$ by lowering it to lower values providing substantial differences to the simple linear relation $\tau = \alpha_0 t$ of the reduced time as a function of the real time t in case of an unchanged initial infection rate α_0 . This has been described and illustrated in detail in Sections 2.2 and 2.3 of our previous publication, reference [10]. The effect of vaccines on the time evolution of pandemic events has been analyzed in detail in reference [36], and is therefore not repeated here.

I am not satisfied with the justification for not including the pre-symptomatic and asymptomatic classes, which both contribute to diseases spread, especially when data is considered. The data considered is only for the confirmed cases, but it is not clear how the authors isolate only these confirmed cases from their model to associate with the data.

Our analysis is based on the reported number of fatalities (deaths) and *not* the number of reported infections, for reasons just highlighted by the referee. The number of fatalities does not suffer from problems concerning the identification of pre-symptomatic and asymptomatic cases.

The initial conditions used to produce some of the figures, e.g., Figures 1 and 2 are not provided meaning it difficult to reproduce the figures.

Actually, all the initial conditions and parameters to reproduce Figures 1 and 2 are listed in Table I. The start time t_0 (column 8), initial condition ηN (column 6) or also η with the size of the population N (column 4), and SIR parameter k (column 5).

The paper is still too technical and more suitable for a more specialized methods or mathematical journal than for a general audience.

Any quantitative modeling of the evolution of pandemics needs some technical mathematical analysis. Most of the technical aspects of our manuscript are arranged in three appendices so that the main text of the manuscript is not overloaded with mathematical formulas. Those that are given are necessary to grasp the logic of the manuscript and to understand the application of monitored data on the number of infected persons. Given (1) that our analysis accounts for any non-pharmaceutical interventions, (2) that it provides an accurate forecast for pandemic wave evolutions in many countries, and (3) that it demonstrates the high accuracy of the SIR model in predicting the evolution of pandemic outbreaks, we think that our manuscript is of high interest for a general audience and readership.

Finally, there are minors typos worth correcting in the manuscript. For example, the authors say "amount of cases" instead of "number of cases" in a number of places including the abstract. They also start the names of months with small letters, e.g., april 2021 in the legend of Table 1 and 11 jan 2021 in line 3 on page 18.

We thank this referee for spotting typos and corrected them in abstract, text and table caption.

Appendix B

Eidgenössische Technische Hochschule Zürich
Swiss Federal Institute of Technology Zurich

Martin Kröger, Prof Dr

Polymer Physics, HCP F 48.2
Leopold-Ruzicka-Weg 4
ETH Zürich
CH-8093 Zürich

Phone +41 44 632 6622
Fax +41 44 632 1076
mk@mat.ethz.ch
www.complexfluids.ethz.ch

Editor Andrew Dunn
Royal Society Open Science

09/08/2021 Zurich

Dear Andrew,

Please find enclosed our revised manuscript entitled "Verification of the accuracy of the SIR model in forecasting based on the improved SIR model with a constant ratio of recovery to infection rate by comparing with monitored second wave data" for re-consideration to Royal Society Open Science. We thank you for your valuable and thoughtful suggestions, to which we fully agree. All changes have been highlighted by blue color (except new data in figures, current data is gray, data at the time of submission and current data separated by a red vertical line). All suggestions are very valuable, and there was no chance to address any of them during preparation of the original manuscript which was received by the Royal Society on 16th January, as you know. We of course did not alter the forecast using newer data during our revision in August, as we are assuming this receipt date will be printed on the manuscript in case of acceptance. All data shown in figures is already available at <https://datadryad.org/stash/dataset/doi:10.5061/dryad.jsxksnogn>, mentioned in the revised Data Availability Statement. The number of references increased, we now cite many more related references that appeared during 2020 and 2021. We changed the title, as requested, and added a new figure 3. Please find below a point-by-point response to all queries.

- (1) *A new title and abstract. The new proposed title is reasonable, the abstract also needs to be edited to remove incorrect predictions about the 2nd wave being the last.*

We changed the title as suggested, and modified the abstract (and many other places). Concerning the part of the manuscript that deals with forecasting, the aim of the revised manuscript is to forecast the dynamics using data from January 2021, and to then also see (during revision) how and why the reported numbers differed from the forecast.

- (2) *Updating figures with current data, making it clear (e.g. with a vertical line or change of colour) when the training data end and where the evaluation point is.*

We added now available data from 12th January onwards to all figures, using another color, and

a vertical red line to separate the two regime. We adjusted the text accordingly. All changes highlighted in blue.

(3) *A discussion of how the training-end and evaluation dates were chosen.*

The training end had not been chosen in any way, but coincides with the day of preparing the manuscript figures (12th January). We submitted our original manuscript on 16th January, i.e., 4 days after preparing the figures (using 11th January data that became publicly available 12th January) to the Royal Society (The manuscript ID was rsif-2021-0046 and Andrew has a copy of the receipt statement; in april it was internally transferred by RS to the RSOS). The preprint, uploaded 5 days after submission, on 21th January, is still available online at <https://www.preprints.org/manuscript/202101.0449/v1>. The newly added verification data was collected friday this week, 20th August. All this is mentioned clearly in the revised manuscript, so that readers understand that the latest available data was used to prepare a forecast, at the time of submission, and that the early forecast had not been adjusted during the last 8 months.

(4) *Checking the writing throughout to avoid anachronisms like "As of today (January 13, 2021)". In my opinion the paper should be as up-to-date as possible at the point of publication.*

We agree. We revised all such phrases and added current data. Because the 2nd wave is over, all data is now final, and the same analysis could be performed on any of the subsequent waves, or to the 2nd wave of the next epidemics.

(5) *A thorough reevaluation of the out of date predictions made in the article such as: "we have shown, the probability for an additional wave is however low in several countries due to the fraction of immune inhabitants at the end of the 2nd wave, irrespective the currently ongoing vaccination efforts." In particular, you should discuss why it is that the prediction from your model fitting has not matched the reality of what has unfolded since the spring. Does it damage the credibility of the method that your prediction of mass immunity preventing further waves was not correct? If not, why not?*

We revised the corresponding statements, added Fig. 3, discuss the important aspect of multiple seeds and covariants of the virus to interpret the data, findings, and conclusions.

Kind regards, in the name of both authors,